# ShapeCrafter: A Recursive Text-Conditioned 3D Shape Generation Model

**Rao Fu**  **Xiao Zhan**  **Yiwen Chen**  **Daniel Ritchie**  **Srinath Sridhar**

Brown University
`rao_fu@brown.edu`
`ivl.brown.edu/projects/shapecrafter`

## Abstract

We present **ShapeCrafter**, a neural network for recursive text-conditioned 3D shape generation. Existing methods that generate text-conditioned 3D shapes consume an entire text prompt to generate a 3D shape in a single step. However, humans tend to describe shapes recursively—we may start with an initial description and progressively add details based on intermediate results. To capture this recursive process, we introduce a method to generate a 3D shape distribution, conditioned on an initial phrase, that gradually evolves as more phrases are added. Since existing datasets are insufficient for training under this approach, we present **Text2Shape++**, a large dataset of 369K shape–text pairs that supports recursive shape generation. To capture local details that are often used to refine shape descriptions, we build upon vector-quantized deep implicit functions that generate a distribution of high-quality shapes. Results show that our method can generate shapes consistent with text descriptions, and shapes evolve gradually as more phrases are added. Our method supports shape editing, extrapolation, and can enable new applications in human–machine collaboration for creative design.

## 1   Introduction

Humans are unique in the animal kingdom in the use of language to describe objects, scenes and events [10]. We use language not only to describe an object's physical and functional attributes, but also to qualify or modify those attribute descriptions. More specifically, we use (linguistic) *recursion* [8]—we may start with an initial description of an object, and progressively add phrases to better describe it or modify it. We use this ability extensively, for instance when describing an object over the phone. The capability to understand recursive natural language descriptions of objects is crucial for the wide accessibility of intelligent robots, computational creativity tools, and online shopping systems.

Driven by recent advances in language modeling [12, 36, 4] and generative image modeling [22], there has been rapid progress in the problem of text-conditioned image generation [55, 44, 57, 58, 59]. For instance, methods such as DALL-E [39, 38] can generate photorealistic images from only text descriptions. This success in the 2D domain together with progress in 3D shape representations [32, 29, 31] has led to interest in text-conditioned *3D shape generation* [50, 40, 20]. However, existing methods have several shortcomings. First, these methods are limited to one-step shape generation and lack the ability to recursively modify or improve generated shapes [13]. Second, unlike in 2D images, there are no readily available large datasets of 3D shape–text pairs. Existing datasets are limited in size due to challenges in manually annotating 3D data and do not support recursive generation [6, 2]. Finally, current methods cannot capture local details that are commonly used in language descriptions.

36th Conference on Neural Information Processing Systems (NeurIPS 2022).

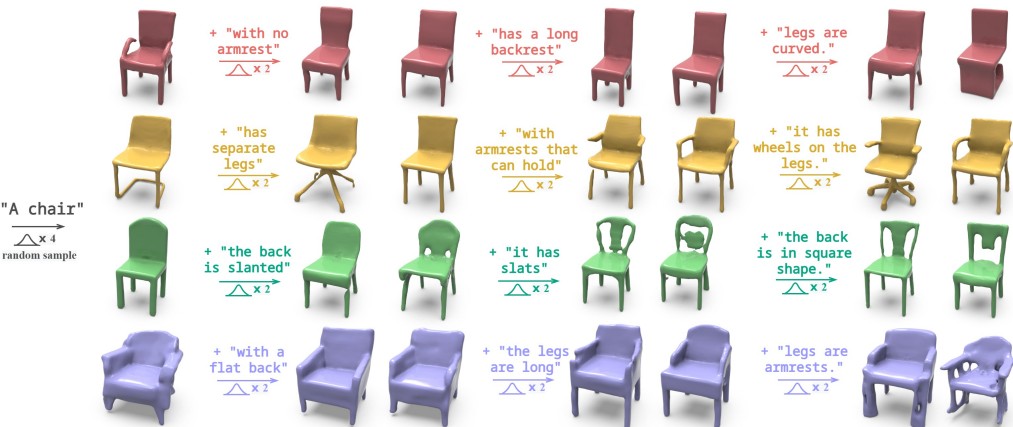

Figure 1: **ShapeCrafter** is a method for recursive text-conditioned 3D shape generation. Given an initial phrase (`A chair`), it generates a shape distribution (4 samples shown in the leftmost column). As more phrases are added, the initial shape is refined (2 samples shown). We can handle long phrase sequences while continuously evolving an initial shape (shape editing for a fixed random seed). Our method also shows extrapolation capabilities (`legs are armrests`, bottom right).

In this paper, we address the above limitations and present a method for **recursive text-conditioned generation of 3D shapes**. Our method, **ShapeCrafter**, consists of a transformer-based [49] neural network architecture that takes an initial text phrase (via BERT [12]) and generates a distribution of latent features over a 3D grid that can be sampled to obtain shapes consistent with the input. Our goal is to capture recursive phrase sequences common in language descriptions (see Figure 1)—as more informative words/phrases are added, our method produces more relevant shape distributions. To achieve this, we design a recursive network architecture that takes the latest phrase input in a sequence and incorporates latent grid features from the previous step to generate refined shapes. The shape distributions produced by our method evolve gradually as more phrases are added, thus enabling new applications in human–machine collaboration for creative design.

To address the limited size of existing datasets, we introduce **Text2Shape++**, a dataset that contains **369K** *many-to-many* mappings of varied-length phrase sequences that map to the same shape. This dataset is derived from Text2Shape [6] that contains 75K one-to-one text-shape pairs without recursive phrase sequences. We introduce a method to transform (see Section 4.1) the Text2Shape dataset into varied-length phrases resulting in a significantly larger dataset. Finally, to learn high-quality shapes, we adopt vector-quantized deep implicit functions (P-VQ-VAE) [56, 31] to better represent local details and generate shape distributions rather than single shapes.

Experiments and results produced by ShapeCrafter demonstrate that we can generate and evolve high-quality 3D shape distributions that are consistent with text inputs. As additional phrases are added to the sequence, the shapes gradually and continuously evolve while retaining shape details generated in the previous steps thus enabling **shape editing** functionality (see Figure 1). Our model can handle long phrase sequences that other methods cannot. Surprisingly, our model demonstrates the capability to extrapolate to novel inputs (*e.g.,* chair legs that look like armrests, see Figure 1 bottom right) unseen during training. Quantitative results show that our shape reconstruction quality is comparable to the state of the art, while our recursive shape generation and editing capabilities are state of the art. To sum up, our contributions are:

- **ShapeCrafter**, a neural network architecture that enables recursive text-conditioned generation of 3D shapes that continuously evolve as phrases are added.
- **Text2Shape++**, a new large dataset of **369K shape–text pairs** that can be used for recursive shape generation tasks.
- A model that captures local shape details corresponding to text descriptions and enables shape editing and extrapolation to novel text descriptions.

## 2  Related Work

In this brief review, we limit our discussion to work on 3D shape representations and 3D shape generation and manipulation. Text-conditioned 2D image generation [24, 39, 38] is outside of the scope of this review, please see [14, 17] for details.

**3D Shape Representation and Generation**: Existing work in 3D shape generation can be categorized based on their underlying representation of choice. Major choices include meshes [16, 54, 51, 15, 11], voxels [9, 26, 21, 6], point clouds [1, 25, 34, 52, 35, 41], neural fields/scene representation networks [30, 42], or deep implicit fields [32, 29, 7]. Meshes and point clouds are useful for simple shapes but struggle with complex details and topology resulting in lower quality results. Voxel grids take up a large amount of memory to preserve details. Implicit fields including SDFs are more popular because they can capture fine-grained details, handle arbitrary topologies, and produce watertight meshes. Since many neural implicits encode the whole shape into a single latent code, they often suffer from "posterior collapse", where the latent space is not fully used, and latents get ignored by the decoder [39, 48]. To combat this, the feature volume can be divided into smaller cells [43, 33]. More recently, vector quantized deep implicit functions (VQ-DIF) [56, 31] have gained popularity. VQ-DIFs encode shapes as series of discrete 2-tuples representing shape features and their position. This allows separation of specific features and effective re-combination in a transformer-based architecture. We adopt this approach in our work due to its high generation quality.

**Text-Conditioned 3D Shape Generation**: Text-conditioned 3D shape synthesis is significantly more challenging than text-conditioned image synthesis due to a lack of large text-shape datasets. Therefore, current methods use large pre-trained image–language embeddings like CLIP [36]. CLIP-conditional shape generation methods [50, 40, 20] associate encoded CLIP text features and CLIP image features obtained by rendering 3D shapes. Alternatively, sequence-to-sequence models [6, 2, 31, 28] are used to match texts and shapes. Unlike our method, none of the above methods can support long phrase sequences or recursive shape generation that controls local shape details.

**Datasets**: Training text-conditioned 3D shape generation models requires supervision in the form of text-shape pairs. Unlike in 2D, obtaining this labeled data is harder in 3D due to the complexity of creating text descriptions for large datasets like ShapeNet [5]. Nonetheless, Text2Shape [6] contains such data for two object classes (chairs, tables) in ShapeNet. In total, they provide 75K pairs (30K for chairs, 40K for tables) which is still insufficient for training models that capture local detail. ShapeGlot [2] provides a (2D) shape discrimination dataset with 94K rendered triplets for shape discrimination. None of these datasets support recursive shape learning. In our work, we provide a method to transform the Text2Shape dataset into a larger dataset of phrase sequence–shape pairs that supports recursive shape generation.

## 3  Background

We first introduce fundamental technical details and justification for our choice of shape representation, text feature extraction, and auto-regressive shape generation.

**Shape Representation**: Neural implicit representations [32] have been shown to capture shapes of arbitrary topology using global latent codes. However, single global latent codes can result in poor local details, so we use grid-based representations [33] which offer better local shape details. In this paper, we use a vector-quantized feature embedding space which represents each shape as a probability distribution of possible tokens in a fixed-size code book.

For each shape $X$, a shape encoder $E_\phi(\cdot)$ encodes its T-SDF (Truncated-Signed Distance Field) into a low dimensional 3D grid feature $E \in g^3 \times D$, where $g$ is the resolution of the 3D grid, and $D$ is the dimension of the 3D grid features. We define a discrete latent space which has $K$ embedding vectors, where each of the vector has dimension $D$. For shape feature $e_i$ at grid position $i$, a vector quantization operation $VQ(\cdot)$ is used to look up its the nearest neighbor $e_i'$ from the $K$ embedding vectors in the discrete latent space. The feature put into the shape decoder $D_\phi(\cdot)$ is the 3D discrete latent feature grid $E' \in g^3 \times D$, with discrete features $e_i'$ at grid locations $i$. Therefore, we have:

$$E = E_\phi(X), E' = VQ(E), X' = D_\phi(E'), \tag{1}$$

where $X'$ is the reconstructed shape, and $\phi$ is the set of learnable parameters of the decoder. As a result, a shape $X$ can be represented as a 3D latent feature index grid $Q \in g^3$, where the number $q_i$ at

each grid location corresponds to a discrete shape feature $e_i'$. To learn this, we use P-VQ-VAE [31], which extracts discrete latent features from volumetric T-SDFs. In our method, we empirically set $K = 512$, $D = 256$, and the 3D shape grid resolution is $g = 8$.

**Text Feature Extraction**: To extract semantic meaning from text inputs, we use BERT [12]. BERT is a bidirectional transformer pretrained on a large web-scale dataset that can be fine-tuned for different tasks. For each input text $I$, we use the first token in last layer of the $\mathbf{BERT_{BASE}}$ model as the text feature embedding $B \in \mathbb{R}^{768}$. In order to have all cells in the 3D feature grid to gather some semantic information from text input, we use a multilayer perceptron network $\Phi(\cdot)$ to project the text embedding feature such that it has the same resolution as the 3D feature grid. We have:

$$C = \Phi(B), \Phi : \mathbb{R}^{786} \to \mathbb{R}^{g^3}, \tag{2}$$

where $C$ denotes the projected text features.

**Autoregressive Generation**: Both the 3D latent feature index grid $Q$ extracted using P-VQ-VAE and the projected text feature $C$ extracted by BERT are in the shape of a 3D grid. Features within the grid are correlated with each other. Our goal is to build a generation model that learns the joint-distribution of the grid while making the code in each grid dependent on previous phrase inputs.

To achieve this, we use a conditional autoregressive model for shape generation. Given grid-wise conditional feature $Z \in g^3 \times H$, where $H$ is the dimension of the conditional feature (we set $H = 512$), the joint distribution of the 3D latent feature index grid $Q$ and the conditional feature $Z$ is formulated as $p(Q|Z) = \prod_{i=1}^{g^3} p(q_i|q_{<i}, Z)$. We use the simplified assumption [31] that this joint distribution is a product of the shape prior, coupled with independent conditional terms $Z$:

$$p(q_i|q_{<i}, Z) = \prod_{i=1}^{g^3} p_\theta(q_i|q_{<i}) \cdot p_\psi(q_i|Z), \tag{3}$$

where $\theta$ and $\psi$ are trainable parameters.

# 4 Recursive Text-conditioned 3D Shape Generation

We now describe the details of how we achieve recursive text-conditioned shape generation. We first introduce our Text2Shape++ dataset in Section 4.1 since the dataset design influences our technical approach. Next, we introduce the "shape set" feature representation we use for recursive generation in Section 4.2. We then introduce the inference procedure of our method in Section 4.3. Finally, we introduce our training strategy that enables recursive generation in Section 4.4.

## 4.1 Text2Shape++: Dataset to Support Recursive Shape Generation

Our goal is to generate 3D shapes recursively from phrase input enabling the gradual evolution of generated 3D shapes. Unfortunately, no existing dataset can support this problem since text–shape pairs are limited to single phrases. Therefore, we construct a new dataset, Text2Shape++, building upon the Text2Shape [6] dataset, which has text–shape pairs of chair and table categories from ShapeNet [5]. Text2Shape++ contains **369K text–shape pairs**, which to our knowledge is the largest dataset of its kind. Each text prompt in the dataset is represented as a phrase sequence, and each phrase sequence corresponds to one or more shapes. On average, each phrase sequence corresponds to 16 models for the chair category, and 25 models for the table category.

**Constructing Text2Shape++**: Given each text prompt in the Text2Shape [6] dataset, we first split the text into sentences using a sentence splitter [19]. For each sentence, we use a constituency parser [23] to parse the sentence in to a constituency tree. There could be many phrase types in a constituency tree—we only consider 18 of them as valid phrase types that contain descriptions such as verbs, nouns, and adjectives. Meanwhile, we also mark the phrases that only contain stop-words[37] (*e.g.,* it, there, . . . ) as invalid. During the parsing procedure, if any child of the current node is not a valid phrase, we stop parsing. Thus, a sentence is parsed into a constituency tree, where all the leaf nodes are valid phrases that contain useful information. For a parsed tree $Y$, we use depth-first search to get all leaf nodes in the tree $I = [i_1, i_2, ..., i_T]$, where $T$ denotes the number of leaves. We use $I_t$ to denote the concatenation of the $1^{st}$ to the $t^{th}$ phrases, and we refer to $I_t$ as a phrase sequence.

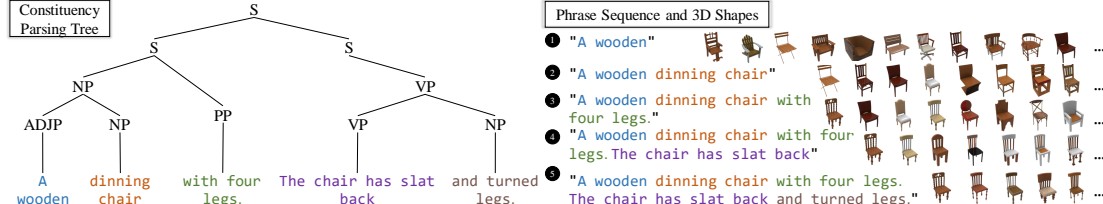

Figure 2: An example from Text2Shape++. Constituency parser [23] annotates a sentence with syntactic structure by decomposing it into phrases. Text2Shape++ contains phrase sequences, and each phrase sequence corresponds to one or more shapes.

To get shapes corresponding to phrase sequences, we calculate the similarity score of all the phrase sequences in our dataset with RoBERTa [27]. We consider two phrase sequences as *similar* if their similarity score is higher than a certain threshold (we empirically chose 0.94). For each phrase sequence, its corresponding shapes include the original Text2Shape dataset pair as well as shapes paired with its *similar* phrase sequences. Therefore, Text2Shape++ is a **many-to-many** text–shape correspondence dataset.

## 4.2 Shape Set Feature Representation.

Since any phrase sequences in Text2Shape++ can have many shapes (*i.e.,*, *shape set*) as their ground truth, we represent the shape set as a probability distribution of the discrete latent space rather a set of deterministic discrete latent feature codes. Recall that we use P-VQ-VAE [31] (see Section 3) to represent a 3D shape $X$ as a 3D discrete latent feature index grid $Q \in g^3$. The discrete latent feature space has $K$ latent features. Therefore, for a shape set that corresponds to a phrase sequence, it can be represented as a probability distribution of latent feature codes $Z \in g^3 \times K$, as illustrated in Fig. 3.

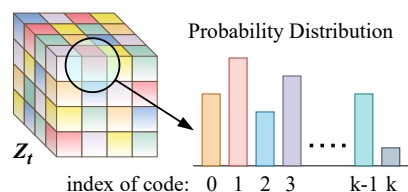

Figure 3: ShapeCrafter learns the probability distribution of latent features for each cell in $Z$.

Formally, each phrase sequence is provided with a shape set $\{X^j\}_{j=1}^M$ and their corresponding similarity scores $\{w^j\}_{j=1}^M$, where $M$ denotes the number of shapes in the shape set. Given shape $X^j$ represented as a 3D discrete latent feature index grid $Q^j$, its probability distribution of latent feature codes $Z$ is deterministic, which can be represented as $Z_{ik} = \mathbb{I}\{k = Q_i\}$, where $i$ denotes grid at position $i$, and $k$ denotes the $k^{th}$ dimension. Given a shape set $\{X^j\}_{j=1}^M$, where each shape $X^j$ is represented by a 3D discrete latent feature index grid $Q^j$, we represent the shape set as a probability distribution by simply weighting the probability of the all the shapes with their similarity score:

$$Z^{set} = \frac{\sum_{j=1}^M w^j * Z^j}{\sum_{j=1}^M w^j}, \tag{4}$$

where $Z^{set}$ is the shape representation of the shape set.

## 4.3 Recursive Shape Generation

We now describe the recursive inference procedure of ShapeCrafter which is also illustrated in Figure 4. ShapeCrafter uses the probability distribution of latent feature codes $Z$ (Section 4.2) as its shape/shape set representation. To achieve text-conditioning, the phrase input is used to change the probability distribution of the latent feature code $Z$. As more phrases are added, the probabilities become narrower resulting in more deterministic shapes.

At time step $t$, ShapeCrafter takes two inputs: text input $i_t$ from the current phrase, and the probability distribution of the latent feature code from the previous time step $Z_{t-1}$. At the first time step, we set $Z_0$ to be $\frac{1}{K}$, where $K = 512$ for each cell. The **BERT**$_{BASE}$ model (Section 3) extracts text features $B_t$ and the multi-layer perceptron network $\Phi(\cdot)$ projects the feature to $C$, which has the same

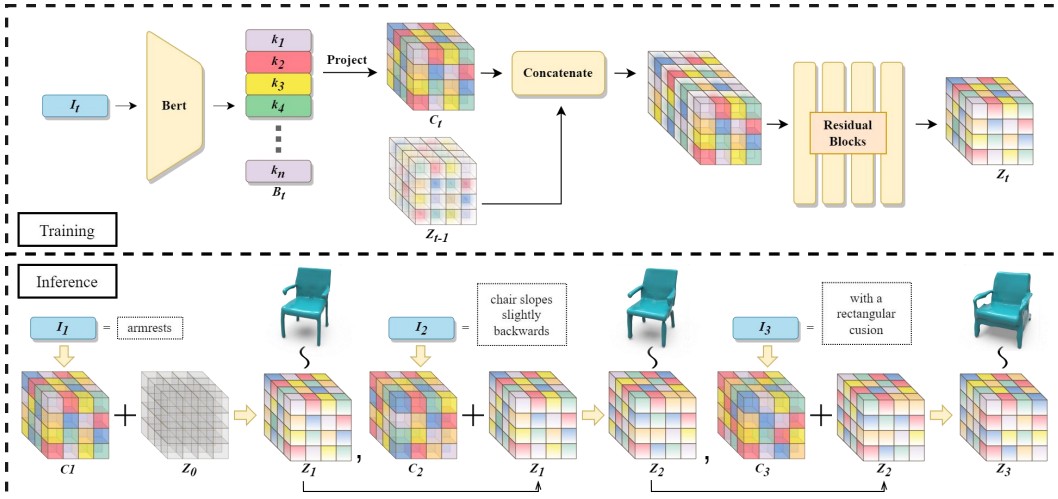

Figure 4: (Top) We take input text phrases and extract semantic features using BERT. These features are projected to a feature grid $C_t$ which is concatenated with the latent feature code distribution $Z_{t-1}$ from the previous time step. Residual blocks $\Psi(\cdot)$ output the feature grid distribution $Z_t$ for the current time step. (2) During step $t$ of inf1.2erence, we combine $C_t$ and $Z_{t-1}$ to obtain $Z_t$ which is sampled to produce 3D shapes.

resolution as the 3D grid features. The text feature $C$ is then concatenated with shape set feature $Z_{t-1}$ along the channel dimension. Several residual blocks $\Psi(\cdot)$ are used to infer the probability distribution of the latent feature code at time $t$, which could be formulated as:

$$Z_t = \Psi([Z_{t-1}, C]), \tag{5}$$

where $[\cdot]$ stands for concatenation. Please see the supplementary document for more details.

The inferred probability distribution $Z_t$ includes information from both text and shape at the previous step. However, the features at each grid are only weakly correlated. Therefore, $Z_t$ is fed to the autoregressive generation model (Section 3) which generates the 3D discrete latent feature index grid $Q$. The VAE decoder $D_\phi$ introduced in Section 3 then decodes the index grid $Q$ to a shape represented by a T-SDF.

At each recursive generation step, we want the generation model to prioritize distinct shape features from the current phrase at $t$. Therefore, we sort the sequence of inputs at the autoregressive step by the difference of the probability distribution at two time steps. We then use a random transformer [45] to generate the index grid sequentially, which could be formulated as:

$$p_t(q_{b_i}|q_{b_{<i}}, Z) = \prod_{i=1}^{N} p_{t\theta}(q_{b_i}|q_{b_{<i}}) \cdot p_{t\Psi}(q_{b_i}|Z), \{b_i\}_{i=1}^{N} = argsort(\|Z_t - Z_{t-1}\|_2^2), \tag{6}$$

where $N = g^3$ is the number of grids, and $argsort(\cdot)$ stands for descending sort.

### 4.4 Training

Five components are involved in training ShapeCrafter: (1) P-VQ-VAE model for shape representation; (2) Auto-regressive model for shape generation; (3) Fine-tuning $\mathbf{BERT}_{BASE}$ model for text feature extraction; (4) Text feature projection model $\Phi(\cdot)$; and (5) Residuals blocks $\Psi(\cdot)$ to extract the final distribution. We first follow the training strategy of AutoSDF [31] to train the P-VQ-VAE model and the autoregressive model, which are trained with both the feature of single shape and the feature of shape set. Then, we jointly train the $\mathbf{BERT}_{BASE}$ model, the projection model , and the residual blocks. Text2Shape++ provides us the dataset to train these three components since it provides phrase sequences $I = [i_1, i_2, ..., i_T]$ and the corresponding shape sets $\{X_t^j\}_{j=1}^{M}$. At time step $t$, we feed in phrase $i_t$ and the shape set probability distribution feature $Z_{t-1}$ corresponding to $I_{t-1}$. Note that $Z_{t-1}$ is taken from Text2Shape++. We receive as output the predicted shape set probability

distribution feature $\hat{Z}_t$. Our loss function consists of reconstruction loss, vector quantization objective, and commitment loss as proposed by van den Oord et al. [47] For the reconstruction loss, we use cross-entropy loss with $Z_t$ as the label. For the vector quantization objective, we randomly sample from the the distribution $Z_t$ and acquire latent feature index $Q_t$ as the target.

# 5 Experiments

In this section, we provide both quantitative and qualitative evaluation of ShapeCrafter on recursive text-conditioned shape generation task. Specifically, we focus on: (1) quantitative evaluation and comparison with AutoSDF [31] to assess the ability of ShapeCrafter to generate high-quality shapes that correspond to text input, (2) quantitative evaluation and comparison with AutoSDF [31] of the ability to handle phrase sequences of varied lengths, (3) qualitative and quantitative evaluation of recursive text-conditioned shape generation, and (4) ablations to evaluate recursive generation in preserving shape details from previous steps. For consistency, we limit our experiments to the chair category, but we show results on the table category in the supplementary document.

**Dataset**: All our experiments use our Text2Shape++ dataset for training (20% held out for validation). For comparisons, we use full sentences from the Text2Shape dataset and phrases from Text2Shape++ dataset for validation. **Metrics**: We use the following three metrics to evaluate text-shape correspondence and shape quality.

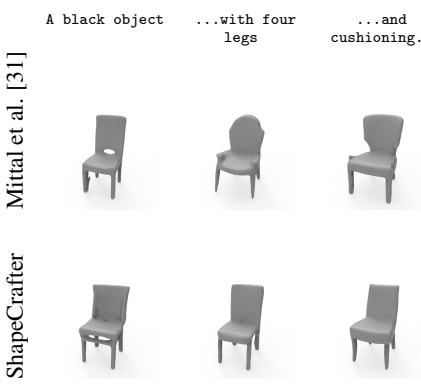

Figure 5: Qualitative comparison with AutoSDF [31]. ShapeCrafter produces sequentially more consistent shapes compared to AutoSDF.

*CLIP-Similarity (CLIP-S).* This metric compares the text-shape correspondence among all the methods as measured by CLIP [36]. We first rendered the generated shape into images from 20 different views. We then use the pretrained CLIP [36] to extract the text feature and images features, and calculate the similarity between the text feature and all of the shape features. We use the maximum of the similarity from the 20 view images as CLIP-Similarity. Even for the same text-shape pair, the CLIP-Similarity value can be different because of the different rendering settings. We normalize the CLIP-Similarity using two standard models, the Stanford bunny[46] and the Utah teapot[3], because we want to reduce the effect of rendering differences. We calculate the cosine similarity between rendered pictures of the Stanford bunny and the word "teapot" and use this score as a lower limit. Then, we take the max cosine similarity between pictures of the Stanford bunny and the word "bunny", and use it as the upper limit.

*ShapeGlot-Confidence (SGLOT-C).* The ShapeGlot-Confidence metric compares the text-shape correspondence between methods. A neural evaluator is trained as proposed in [2] to distinguish the target shape from distractors given the specified text. The trained neural evaluator achieves an 83% accuracy on this binary classification task with the Shapeglot[2] dataset. ShapeGlot-Confidence informs us which shape most closely corresponds to the text input among all shapes generated by the methods that we are comparing. We use this as a proxy for a perceptual study.

*FID.* We use the Frechet Inception Distance (**FID**) metric [18] to evaluate the quality of the generated shapes. For shape feature extraction, we use the voxel encoder in [53] which is trained on the ShapeNet classification task[5].

## 5.1 Comparisons

We evaluate and compare the performance of ShapeCrafter on text-conditioned shape generation [31]. For this experiment, we use single step generation for AutoSDF using complete prompts from the Text2Shape [6] test set (since AutoSDF is not designed for recursive generation). Our model generates recursively by taking the sentence as input, parsing it into phrases, and generating according to the parsed phrases recursively. Table 1 illustrates the performance of ShapeCrafter and AutoSDF[31] on

this task. We outperform AutoSDF on the CLIP-Similarity score and the Shapeglot-Confidence score, which means ShapeCrafter is able to generate shapes that are visually more similar to the text input. For shape quality evaluation, our model has a lower FID score, indicating that we can capture details better. This experiment also shows that the recursive generation strategy is useful, compared to the single step strategy.

In Figure 5, we also compare the performance between AutoSDF[31] and ShapeCrafter qualitatively. We visualize the shapes generated by phrases from Text2Shape++. Clearly, comparing with AutoSDF, the generated results of ShapeCrafter is more sequentially consistent.

Table 1: AutoSDF and ShapeCrafter on text-conditioned generation. Compared to AutoSDF, ShapeCrafter performs better on CLIP-S, SGLOT-C, and FID, which indicates that it provides better text-shape correspondence and shape quality.

| Metric | CLIP-S ↑ | SGLOT-C ↑ | FID ↓ |
|---|---|---|---|
| Mittal et al. [31] | 48.92 | 0.46 | 18.45 |
| ShapeCrafter (Ours) | **52.43** | **0.53** | **16.36** |

## 5.2 Phrase Sequence Length

We are also interested in the ability of ShapeCrafter on generating text inputs of various lengths. We divide the texts inputs into three levels by the number of phrases they contain. All text inputs are categorized into three tiers: texts that contain $1-2$, $2-4$, and more than $4$ phrases. Table 2 compares the performance of AutoSDF and ShapeCrafter on text inputs of various lengths. Overall, the results from these three metrics indicate that ShapeCrafter generates shapes that are more consistent with longer text inputs than AutoSDF without compromising generation quality. Specifically, for FID, the shape quality of AutoSDF decreases as the text input grows longer, while the shape quality of ShapeCrafter remains stable. For CLIP-Similarity, the performance of ShapeCrafter is comparable with AutoSDF for shorter texts, and ShapeCrafter is better than AutoSDF at generating shapes that closely agree with medium-length and long texts. For Shapeglot-Confidence, AutoSDF performs better with shorter/medium-length text inputs, but ShapeCrafter performs better with longer inputs. The result shows that recursive generation strategy outperforms single step strategy in longer text inputs.

Table 2: AutoSDF and ShapeCrafter on recursive text-conditioned shape generation. ShapeCrafter uses a recursive strategy and it performs better with longer inputs.

| # Phrases | [1, 2] | | | (2, 4] | | | (4, +∞) | | |
|---|---|---|---|---|---|---|---|---|---|
| | CLIP-S ↑ | SGLOT-C ↑ | FID ↓ | CLIP-S ↑ | SGLOT-C ↑ | FID ↓ | CLIP-S ↑ | SGLOT-C ↑ | FID ↓ |
| Mittal et al. | **45.72** | **0.53** | 18.13 | 45.27 | 0.42 | 20.33 | 55.77 | 0.43 | 22.47 |
| ShapeCrafter | **45.72** | 0.47 | **17.40** | **53.38** | **0.58** | **16.44** | **58.18** | **0.57** | **16.89** |

## 5.3 Recursive Text-conditioned Shape Generation

We evaluate the performance of ShapeCrafter on recursive text-conditioned shape generation task qualitatively. Fig. 6 shows the shape generated at each time step. Shapes rendered in the same color are generated from the same phrase sequences. This figure provides examples that demonstrate ShapeCrafter has the ability to (1) generate high-quality shapes *(row 1)*, (2) modify the global structure of shapes *(row 2)*, (3) change part-level attributes *(row 3-5)* including part addition, part removal, and part replacement, (4) modify the local details of the shape *(row 6)*, (5) generate from long phrase sequences *(row 7 example 1)*, (6) generate novel shapes *(row 7 example 2)*. The above properties shows ShapeCrafter performs well in text-conditioned shape generation and text-conditioned shape editing.

Table 3 shows how the probability distribution $Z$ and generated shape changes with the number of text phrases increases. For a text phrase sequence, we calculate (a) the mean entropy(H) of the per-grid-cell discrete probability distributions $H = -\sum_{i=1}^{g^3} \sum_{k=1}^{K} z_{ik} \log(z_{ik})/g^3$, and (b) the mean Chamfer Distance(CD) among $8$ randomly sampled shapes.

The table shows that the entropy and the mean Chamfer Distance decrease with more phrases. This indicates that as more phrases are added, the probabilities become narrower resulting in more deterministic shapes.

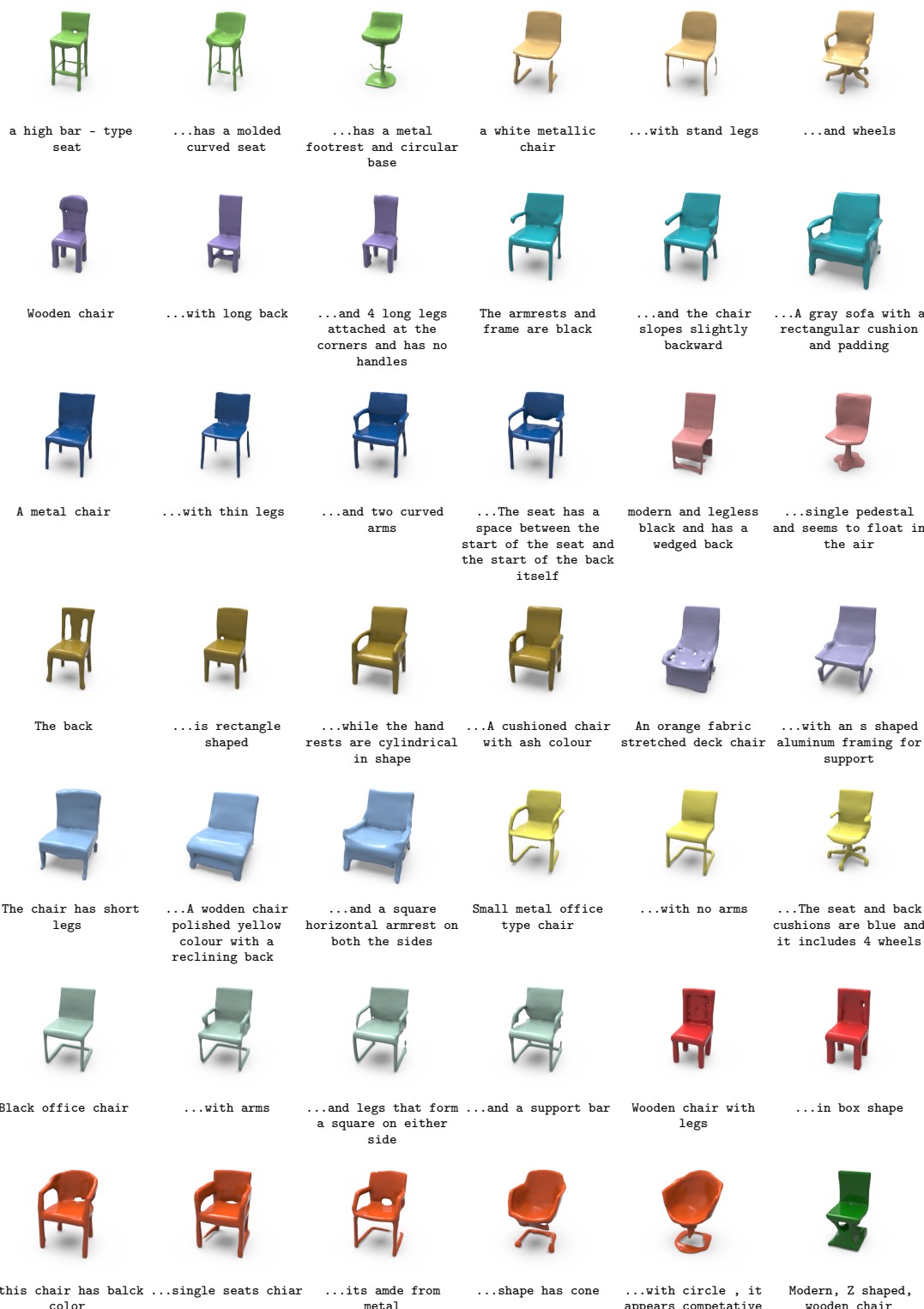

Figure 6: Qualitative results produced by ShapeCrafter. Each unique color shows results generated recursively from our model using the same seed to sample the shape distribution. Our method demonstrates consistent and gradual evolution of shape enabling us to simulate shape editing.

Table 3: The Entropy(H) of the probability distribtuion and the Chamfer Distance(CD) between the generated shapes at sentence with different number of phrases. The probabilities narrow as more phrases are added, resulting in more deterministic shapes.

| # Phrases | 1 | $[2, 4)$ | $[4, 8)$ | $[8, +\infty)$ |
|-----------|-------|----------|----------|----------------|
| H | 0.423 | 0.341 | 0.294 | 0.267 |
| CD | 0.0861 | 0.0698 | 0.0359 | 0.0261 |

## 5.4 Ablations

We design ablations to evaluate whether: (1) the conditional training strategy improves text-shape correspondence, (2) the random transformer improves shape quality; and (3) reordering transformer inputs preserves shape details. To prove the effectiveness of conditional training, we design a baseline removing the residual blocks $\Psi(\cdot)$ and only generating shape features recursively with AutoSDF [31]. Specifically, at time step $t$, we multiply the probability distribution $C_t$ generated from the text input with the

Table 4: Ablation studies shows the effectiveness of the conditional training, the random transformer and the reordered random transformer sequence.

| Metric | CLIP-S $\uparrow$ | FID $\downarrow$ |
|--------|---------|---------|
| w/o condition | 49.53 | 20.22 |
| w/o transformer | 49.29 | 19.83 |
| w/o reorder | 42.47 | 16.67 |
| Ours | **52.43** | **16.36** |

probability distribution generated from the random transformer at $t - 1$, and we use the product as the conditional distribution of the random transformer at $t$. We refer to this baseline as *w/o condition*. To prove that the random transformer helps improve the quality of the generated shapes, we design a baseline *w/o transformer*, where we sample shapes from the probability distribution generated by the residual blocks $\Psi(\cdot)$. To prove the effectiveness of reordering the input sequence in the random transformer (refer to section 4.3), we design another baseline where the transformer generates features in a random sequence. We refer to this baseline as *w/o reorder*. The performance of these baselines are illustrated in Table. 4. The performance drops without these components, which proves the effectiveness of our design choices.

## 6  Conclusion

In this paper, we present ShapeCrafter, a method for recursive text-conditioned 3D shape generation. Different from previous single-step methods, our focus is on generating 3D shape distributions that can be gradually evolved to capture semantics described in phrase sequences. We also presented a method to transform an existing dataset to support recursive shape generation tasks to produce a larger dataset called Text2Shape++ with 369K shape–text pairs. Results show that our method produces high-quality shapes while being able to handle long phrase sequences, support shape editing, and extrapolate to shapes unseen during training.

**Limitations**: Our method has several notable limitations. First, ShapeCrafter is limited to shape generation only and cannot handle appearance attributes common in language descriptions. Limitations of the Text2Shape dataset extend to our method—we only support two shape categories: tables and chairs. Empirically, we observe that our method demonstrates elements of "part assembly", but cannot handle large-scale deformations. The proposed recursive shape generation method is not reversible. For example, given an original shape with inputs "with armrest" and "without armrest" applied sequentially, the generated shape may not be identical to the original shape. Another noticeable problem in the text-conditioned shape editing community is that there lacks a common metric to evaluate the accuracy of shape editing. We believe that we have taken a step in the direction towards addressing the above challenges. From a **societal impact** perspective, more care is needed to ensure that our dataset represents diverse shape instances to avoid bias.

**Acknowledgements**   This research was supported by AFOSR grant FA9550-21-1-0214, NSF CNS-2038897, and the Google Research Scholar Program. Daniel Ritchie is an advisor to Geopipe and owns equity in the company. Geopipe is a start-up that is developing 3D technology to build immersive virtual copies of the real world with applications in various fields, including games and architecture.

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
