# OpenReview forum: "ShapeCrafter: A Recursive Text-Conditioned 3D Shape Generation Model"
_NeurIPS.cc/2022/Conference — NeurIPS 2022 Accept_

### Official Review · Reviewer_BoQp · 2022-06-27

**Rating:** 4
**Confidence:** 5
**Soundness:** 2 fair
**Presentation:** 3 good
**Contribution:** 2 fair

**Summary:**

The paper discusses a method to sequentially generate a chair from a sequence of utterances provided by the user. To do so the paper first cleans the Text2Shape++ dataset by applying linguistic parsing. Their model consists extracting text features from text, that are then projected to a volumetric grid and concatenated to the previously generated volumetric features. Then this tensor is used to generate an SDF to represent the shape. The paper shows examples of a couple of sequential edits.

I wonder why authors used Recursive in the title, since it has a connotation that it uses a recurrent NN, or any other recurrent process. Here instead it's sequential, with not too many steps presented in the paper.

**Questions:**

I'd like the authors to comment on the questions listed in the weaknesses section. I believe the paper is a good piece of work, that might be not quite ready for a publication

**Limitations:**

The limitation section is satisfactory

**Strengths And Weaknesses:**

Strength:

S1. The paper definitely tackles an interesting project, as far as I'm aware, this is the first of one of the first papers in this domain.
S2. Their architecture is somewhat new, somewhat extended from ShapeFormer.
S3. Written very well, thank you, very easy to read and follow.

Weaknesses:

W1. **Results**. The paper report standard metrics that evaluate how well the shape corresponds to the text. The paper, however, does something different, they propose a *sequential* generation paper. While ShapeGlot-Confidence or Clip-Score are necessary, they're by no means sufficient. How can we be sure that sequential generation happened. If you say "a chair with straight legs" and then add "curved thin legs" and then add "with a hole in the seat", we'd like to be sure that only the these attributes change, and only when asked. The paper doesn't provide a metric or evaluation protocol for that. A further interesting metric to consider is to evaluate that other parts of the chair stay the same or marginally change while attributes are added. The paper doesn't provide that too. Such metrics probably don't exist, but I believe the responsibility of coming up with a reasonable evaluation protocol bears with the authors and if done well can be listed as a contribution.

These metric are very important for such a method, because, if we look at Fig 3, we see that changes to not mentioned parts of the chair are significant. For example in the bottom row "a black object" gives you a chair with four legs, "...with four legs" gives another chair with four legs, "...and cushioning" gives another chair with four legs and slightly changed geometry. I think any of these three chairs perfectly satisfies a description "a black object with four legs and cushioning" I cannot tell whether the final variant is the best. Also it's not clear why "...with cushioning" makes the legs curvy. Inconsistency of sequential updates is also the case in examples in Fig.4.

W2. **Adding vs Removing attributes**. Currently the utterances are additive, ie we add attributes to the chair, they might be conflicting such as "stand legs" and "wheels". But I wonder if it's possible to say mutually exclusive things, such as "a char with four legs", then "a chair with one metal leg" and then "a chair with four legs". OR "a chair with armrests" and then "a chair without armrests". I wonder if there is a metric to compute, when you do subtract attributes you should return to the exactly previous chair, perhaps it's even a decent loss function...

W3. **Generality**. In its current form the paper operates only on chairs, for whatever reason they add attributes of materials to the text, while not being able to render "black, metal, wooded" and so on. Have you thought about other categories? The paper title is quite generic, while implementation is category specific. It would be good to have one model to generate multiple object categories.

---

> ### Author Response · Authors · 2022-08-01
> **Response to Reviewer BoQp - Strengths**
>
> We thank the reviewer for the careful reviews and constructive suggestions. To recap, recursive text-conditioned shape generation is a novel task, which has not been investigated by the community before. We are glad to see all reviewers appreciating our work in the following aspects: (1) the task is novel and useful (e81L, BoQp); (2) the dataset is constructed in an automatic way (Bwe4), which is useful and unique (e81L, Bwe4); (3) the shape set representation as a probability distribution is interesting (Bwe4); (4) the method that supports probabilistic generation is new (Bwe4, BoQp); (5) the method is effective and generates better quality shapes than previous works (e81L); (6) apply the method to a real-world demo (Bwe4); (7) the paper is well-written (BoQp).
>
> In this response, we focus on addressing questions, providing evidence, and addressing factual errors.

---

> > ### Author Response · Authors · 2022-08-01
> > **Response to Reviewer BoQp's Concern - Generality, Adding vs Removing attributes, Title**
> >
> > > **Generality. In its current form the paper operates only on chairs, for whatever reason they add attributes of materials to the text, while not being able to render "black, metal, wooded" and so on. Have you thought about other categories? The paper title is quite generic, while implementation is category specific. It would be good to have one model to generate multiple object categories.**
> >
> > A: *We demonstrate results on table category in section 2 of the supplementary, which shows that our method is able to generalize to other categories*. We found the shape geometry changes with materials. For example, in Fig 4, 'metal' chairs have thinner legs, and 'wooden' chairs have thicker legs. This indicates that the geometry is correlated with materials, so we left the material descriptions unchanged.
> >
> > We consider that our major applications (shape generation, shape editing) are usually category-specific [1-5], so we choose to investigate a category-specific model.  We will add this point to the limitation section.
> >
> > [1] StructureNet: Hierarchical Graph Networks for 3D Shape Generation (Siggraph Asia 2019)
> >
> > [2] PolyGen: An Autoregressive Generative Model of 3D Meshes (PMLR 2020)
> >
> > [3] SP-GAN: Sphere-Guided 3D Shape Generation and Manipulation (SIGGRAPH 2021)
> >
> > [4] DSG-Net: Learning Disentangled Structure and Geometry for 3D Shape Generation (ACM TOG 2022)
> >
> > [5] A Revisit of Shape Editing Techniques: From the Geometric to the Neural Viewpoint (CVM 2021)
> >
> > > **Adding vs Removing attributes. Currently the utterances are additive, ie we add attributes to the chair, they might be conflicting such as "stand legs" and "wheels". But I wonder if it's possible to say mutually exclusive things, such as "a char with four legs", then "a chair with one metal leg" and then "a chair with four legs". OR "a chair with armrests" and then "a chair without armrests". I wonder if there is a metric to compute, when you do subtract attributes you should return to the exactly previous chair, perhaps it's even a decent loss function...**
> >
> > A: The reviewer brings up an interesting point about 'conflicts' in the input text prompt: either a conflicting attribute of the same part or conflicting existence of a part. While none of the previous text-conditioned shape generation papers report any editing with such 'conflicted' text prompts, we indeed tried editing with such texts during our experiments. As a result, the model can sometimes generate a novel shape which is a hybrid of the conflicting attributes. For example, if the inputs are 'standard legs' and 'wheels', the model can generate a shape that looks like the last example in Figure 1 row 2, where the shape has wheels on standard legs. Most of the time, the model tends to choose one of the attributes. This could be due to our model representing the shape as a probability distribution. Therefore, when the model is only conditioned on one attribute, the probability distribution is more narrow. When the model is conditioned on more than one attribute, the probability distribution widens out. We will add visualizations of editing with 'conflicted' text prompts in the supplementary material.
> >
> > > **I wonder why authors used Recursive in the title, since it has a connotation that it uses a recurrent NN, or any other recurrent process. Here instead it's sequential, with not too many steps presented in the paper.**
> >
> > A: We differentiate the concept of 'recursive' and 'recurrent': 'recurrent' models is a subset of 'recursive' models, all of which act in a recursive manner, but the 'recurrent' models have explicit 'time-stamp' in the model. Our paper uses 'recursive' for two aspects: 1. The output of the model at timestep $t>0$ depends on the output at the last timestep. The inference process is repeated in the same manner at each time step. 2. From a language description perspective, a sentence can be parsed into a recursive phrase tree, so we keep the 'recursive' concept here. We also cite [7] referring to this use of 'recursive'.

---

> > ### Author Response · Authors · 2022-08-01
> > **Response to Reviewer BoQp's Concern - Results**
> >
> > > **The paper report standard metrics that evaluate how well the shape corresponds to the text. The paper, however, does something different, they propose a sequential generation paper. While ShapeGlot-Confidence or Clip-Score are necessary, they're by no means sufficient. How can we be sure that sequential generation happened. If you say "a chair with straight legs" and then add "curved thin legs" and then add "with a hole in the seat", we'd like to be sure that only the these attributes change, and only when asked. The paper doesn't provide a metric or evaluation protocol for that. A further interesting metric to consider is to evaluate that other parts of the chair stay the same or marginally change while attributes are added. The paper doesn't provide that too. Such metrics probably don't exist, but I believe the responsibility of coming up with a reasonable evaluation protocol bears with the authors and if done well can be listed as a contribution.**
> >
> > A: This is a fair critique, thank you. We did a literature review on image editing and shape editing and found that there is no commonly-used metric except for perceptual studies to measure local geometry changes. However, we have designed a metric for our problem and are happy to include this in the camera-ready version. We calculate the probability distribution difference of shape feature at a grid cell at two consecutive time step with: $\text{diff} = \max(z_{t,k}-z_{t-1,k}) \in [0, 1]$, where $k \in \{1, 2, ..., K\}$, and $K$ is the number of codes in the codebook. At two consecutive time steps, we report the ratio of grid cells whose probability distribution difference is smaller than a threshold $\tau$ among all $8^3$ grid cells. In the table below, we evaluate the local geometry change with *the percentage of distribution difference metric* and compare with AutoSDF.
> >
> > | $\tau$ | 1e-10 | 1e-9 | 1e-8 | 1e-7 | 1e-6 |
> > | ----- | ----- | ----- | ----- | ----- | ----- |
> > | AutoSDF |   1.70 | 4.31 |9.65 | 17.1 | 25.08 |
> > | Ours    |  6.98 | 11.96 | 18.03 | 24.17 | 29.56 |
> >
> > As we see, ShapeCrafter has a lower percentage of grid cells changing from step to step than AutoSDF. It shows that recursive generation changes localized regions. We acknowledge limitations of this metric -- the changed region does not necessarily semantically corresponds to the change in the input text; but this is a hard problem. Instead, we visualize the distribution difference in Fig 3 of the supplementary (the arrows between the 3rd-4th rows are linked incorrectly; we will fix them in the final version). The distribution is only changed locally by text inputs. The first phrase changes the grid cells where a chair could exist, the second phrase changes the back and the seat part, and the third phrase changes the armrest part. This shows that the changed region corresponds to the input text.
> >
> > > **These metric are very important for such a method, because, if we look at Fig 3, we see that changes to not mentioned parts of the chair are significant. For example in the bottom row "a black object" gives you a chair with four legs, "...with four legs" gives another chair with four legs, "...and cushioning" gives another chair with four legs and slightly changed geometry. I think any of these three chairs perfectly satisfies a description "a black object with four legs and cushioning" I cannot tell whether the final variant is the best. Also it's not clear why "...with cushioning" makes the legs curvy. Inconsistency of sequential updates is also the case in examples in Fig.4.**
> >
> > A: The metric above and the Fig 3 of the supplementary show that the probability distribution changes at the regions that are semantically corresponding to the text input. The inconsistency of generated shapes mainly results from the sampling process and the conditional random transformer. Fig 4[c] in the supplementary material provides an example. The first rows are the shapes sampled from the output of the residual blocks $\Psi()$, and the second rows are sampled from the output of the random transformer. The seats and the backs look consistent before the transformer but become less consistent after the transformer. This is because the probability of the next token is the product of the transformer output and the $\Psi()$ output (see equation 3). This means that the shape of each grid cell at a different time step is not deterministic, but it has a higher probability to be consistent. If most of the grid cells are consistent, the shapes tend to look similar. The table above shows that our recursive generation method can preserve sequential consistency better than random generation.
> > Also, as the probability of the first token is majorly decided by $\Psi()$, the random transformer tends to generate a shape similar to the first observed tokens. As we shuffled the input order by the difference of $Z_t$ with the previous step, the grid cells with a smaller difference will become less similar after the transformer.

---

> ### Author Response · Authors · 2022-08-08
> **Looking forward to hearing the responses from reviewer BoQp**
>
> We thank the reviewer for the previous careful reviews and suggestions. We also would like to hear your further comments and suggestions. Please leave a comment and let us know if there are any other questions.

---

> ### Author Response · Authors · 2022-08-09
> **Looking forward to hearing the responses from reviewer BoQp**
>
> Dear reviewer, thanks for appreciating the novel task and the architecture in our paper, and we hope our answer addresses your questions and concerns. We are looking forward to hearing back from you. Please let us know if you have any other questions or concerns about the paper.

---

### Official Review · Reviewer_Bwe4 · 2022-07-09

**Rating:** 5
**Confidence:** 4
**Soundness:** 3 good
**Presentation:** 3 good
**Contribution:** 2 fair

**Summary:**

The paper presents a neural network for recursive text-conditioned 3D shape generation --- progressively updating the generated shape by a series of input phrases. To achieve this, the proposed method is built on top of vector-quantized deep implicit functions (as the shape representation) such that a set of possible shapes can be described as a distribution over 3D grids, which can then be learned using an autoregressive model. Another contribution of this paper is a enhanced dataset, Text2Shape++, that provides many-to-many shape-text pairs. The proposed method largely relies on such dataset.

The results demonstrated that the method can generate shapes from text input, and the shapes evolve gradually as more short phrases are added. The authors also include a realtime demo where the user describes a shape by speech and the generation results are shown in the screen.


**Questions:**

Questions and suggestions:
- Line 202-205, the sorting is a bit counter-intuitive to me. Wouldn't this sort operation make the order inconsistent among the dataset? My first thought is that such inconsistent order would hurt the training. I suggest elaborating more on the random transformer, which seems to be a key choice here.
- Line 237-240, why is it necessary to normalize clip-similarity score? Cosine similarity by itself already has limited value range.
- For all loss functions described in Sec. 4.4, I suggest including their mathematical formulations, at least in the supplementary document, to make the paper self-contained.

Typos:
- Line 139, "since since" -> "since".
- Line 148, "... is the largest such dataset" -> "... is the largest dataset of its kind".
- Line 177, $X_j$ -> $X^j$.


**Limitations:**

Limitations are well discussed and several failure cases are presented in the supplementary document. Negative social impact is also addressed.

**Strengths And Weaknesses:**

Strengths:
- It is an interesting idea to represent a set of possible shapes as a probability distribution over latent feature codes. With an autoregressive generator, this design naturally supports probabilistic generation (i.e., capable of producing different shapes from the same input). This is a different and novel approach compared to previous text2shape methods.
- The Text2Shape++ dataset, even though constructed for a particular scenario, is a good contribution to the community. I like the way that the authors construct the dataset, i.e., use principled natural language techniques such as constituency parser and phrase similarity.
- For an application paper, a real-world demo is a bonus. I appreciate the good effort by the authors in making such demo.

Weaknesses:
- My main concern is about the result quality and evaluation.
    - Poor visual quality. Many of the generated shapes (i.e., those in Fig.1, Fig.3, Fig.4) have noticable artifacts, e.g., holes, disconnected or floating parts. Given the visual quality of the demonstrated examples, I would doubt the overall result quality.
    - Missing comparison to previous SOTA text2shape methods, especially [1]. The compared method, AutoSDF is not especially designed for text-driven shape generation. Another fact that I noticed is that this work ignores the shape color (so does autoSDF), which is an important information for many text descriptions and is considered in previous text2shape literatures [1][2]. I'd like to hear the reason that the surface color is not considered. Clearly the dataset itself contains colored shapes.
    - Missing support for the made claim. Specifically, line 187-188, "As more phrases are added, the probabilities become narrower resulting in more deterministic shapes.". I think this should be a key feature of the method, but somehow it is not validated in the experiments.
- Another minor concern is the application scope or the motivation for recursive text-conditioned generation. From my perspective, the advantage of text-driven 3D shape generation is to quickly produce a base/rough shape. People (modeling experts or ordinary users) then start from that shape for further detailed editing, from where precise control is more important. However, from the presented results, I'm not convinced that recursive text input can provide such necessary precise control (as text itself is vague), compared to traditional 3D shape editing techniques.

Given that my concerns outweigh the positive contribution of the paper, I'm inclined to rejection at this point.

[1] Zhengzhe Liu, Yi Wang, Xiaojuan Qi, and Chi-Wing Fu. Towards implicit text-guided 3d shape generation. CVPR 2022.
[2] Kevin Chen, Christopher B Choy, Manolis Savva, Angel X Chang, Thomas Funkhouser, and Silvio Savarese. Text2shape: Generating shapes from natural language by learning joint embeddings. ACCV 2018.

__Update:__

After reading the rebuttal and other reviews, I feel more positive about this paper as some of my major concerns are well addressed (missing comparison, lacking support for the claim). Thus I raise my rating from 4 to 5.

---

> ### Author Response · Authors · 2022-08-01
> **Response to Reviewer Bwe4**
>
> We thank the reviewer for their review. To recap, recursive text-conditioned shape generation is a novel task, which has not been investigated by the community before. We are glad to see all reviewers appreciating our work in the following aspects: (1) the task is novel and useful (e81L, BoQp); (2) the dataset is constructed in an automatic way (Bwe4), which is useful and unique (e81L, Bwe4); (3) the shape set representation as a probability distribution is interesting (Bwe4); (4) the method that supports probabilistic generation is new (Bwe4, BoQp); (5) the method is effective and generates better quality shapes than previous works (e81L); (6) apply the method to a real-world demo (Bwe4); (7) the paper is well-written (BoQp).
>
> In this response, we focus on addressing questions, providing evidence, and addressing factual errors in the review. **We hope that the reviewer can re-evaluate their rating in light of the evidence we present below to address concerns and factual errors in their review**.

---

> > ### Author Response · Authors · 2022-08-01
> > **Response to Reviewer Bwe4 - Other questions**
> >
> > We thank the reviewer for appreciating the probability represetation we use, the probabilistic model we adopt, the dataset we construct, and the live demo we presented. We hope that the reviewer will reconsider their score given the data and facts we have presented that the shape quality is comparable with the state-of-the-art, the missing claim is supported, and recursive text-conditioned generation is a useful task.

---

> > ### Author Response · Authors · 2022-08-01
> > **Response to Reviewer Bwe4 - Other questions**
> >
> > ### Answers to the other questions
> > > **Another fact that I noticed is that this work ignores the shape color (so does autoSDF), which is an important information for many text descriptions and is considered in previous text2shape literatures [1][2]. I'd like to hear the reason that the surface color is not considered. Clearly the dataset itself contains colored shapes.**
> >
> > A: We clearly acknowledged this limitation in the paper. There exists numerous papers on shape generation that solely focus on generating geometry, which is still a very valuable problem to investigate (see references [1-4] below). Also, the ShapeNet Core V1 dataset, which includes Text2Shape, contains poor-quality textures. It is commonly understood that ShapeNet is useful for shapes, but not for high-quality texture generation. We keep the appearance descriptions because they are correlated with shape geometry. For example, in Fig 4., wooden chairs have thicker legs, and metal chairs have thinner legs. Therefore, in this new generation task, we generate geometry only and keep the appearance description. We strongly believe that investigating shape in itself is a valuable contribution to make.
> >
> > [1] StructureNet: Hierarchical Graph Networks for 3D Shape Generation (Siggraph Asia 2019)
> >
> > [2] PolyGen: An Autoregressive Generative Model of 3D Meshes (PMLR 2020)
> >
> > [3] SP-GAN: Sphere-Guided 3D Shape Generation and Manipulation (SIGGRAPH 2021)
> >
> > [4] DSG-Net: Learning Disentangled Structure and Geometry for 3D Shape Generation (ACM TOG 2022)
> >
> > > **Another minor concern is the application scope or the motivation for recursive text-conditioned generation. From my perspective, the advantage of text-driven 3D shape generation is to quickly produce a base/rough shape. People (modeling experts or ordinary users) then start from that shape for further detailed editing, from where precise control is more important. However, from the presented results, I'm not convinced that recursive text input can provide such necessary precise control (as text itself is vague), compared to traditional 3D shape editing techniques.**
> >
> > A: We disagree -- recursive text-conditioned generation has many applications beyond the ones mentioned by the reviewer: (1) For 3D design, novice users (no 3D experience) can select shapes based on recursive generation and customize shapes based on aesthetics (e.g., make the backrest smaller). (2) For human-robot interaction, recursive generation enables users to provide feedback to previously generated shapes. (3) We also show our advantage over one-shot generation in Fig 3: we can change a single attribute while keeping the other attributes similar. In Figure 4, examples show that text-conditioned editing can control local details, like r3c4 (change the back slightly backward), r4c4 (connect legs), and r5c5 (change the leg shape). The vagueness of language makes recursive text-conditioned generation important because it enables users to obtain feedback and refine the shape.
> >
> >
> > > **Line 202-205, the sorting is a bit counter-intuitive to me. Wouldn't this sort operation make the order inconsistent among the dataset? My first thought is that such inconsistent order would hurt the training. I suggest elaborating more on the random transformer, which seems to be a key choice here.**
> >
> > A: By "random transformer," we mean an order-invariant transformer. The original transformer assumes a fixed ordering of input tokens, which means the output token $z_i$ at position $i$ depepends on the output of all the tokens $z_{<i}$ smaller than position $i$. That is $p(Z) = \sum_{i=1}^{g^3}p(z_{i}|z_{<i})$. However, conditional generation tasks such as image composition [43], video synthesis [43] and shape completion [30] may not guarantee a fixed ordering of input tokens. Take shape completion, for example: a partial shape would be represented as known tokens at random position $z_{b_{<i}}$. The next token $z_{b_{i}}$ is generated from all of the previous known tokens $z_{b_{<i}}$. Therefore, the output is not spatially sequential. That is, $p(Z) = \sum_{i=1}^{g^3}p(z_{b_{i}}|z_{b_{<i}})$. To achieve this goal, the transformer is trained with randomly shuffled input order, making the random transformer be robust to order variation.
> >
> > > **Line 237-240, why is it necessary to normalize clip-similarity score? Cosine similarity by itself already has limited value range.**
> >
> > A: Even for the same shape-text pair, the CLIP-Similarity value can be different because of the different rendering settings. We normalize the CLIP-Similarity using two standard models because we want to even out the rendering factors. Also, we don't want the readers of the paper to care too much about the absolute value of the cosine similarity because only the relative value matters.
> >
> > > **Loss functions. Typos.**
> >
> > A: Thanks for the suggestion. We will add the formulations of the loss functions to the camera-ready version. We will revise all of the typos in the paper.

---

> > ### Author Response · Authors · 2022-08-01
> > **Response to Reviewer Bwe4 - Main concerns**
> >
> > ### Answers to the main concerns
> > > **Missing comparison to previous SOTA text2shape methods, especially [1]. The compared method, AutoSDF is not especially designed for text-driven shape generation.**
> >
> > A: This statement is factually incorrect because we compared the performance of our method with [1] in Tables 1 & 2 of the supplementary (**Note: We compared with this method even though it was not published at the time of submission**). We apologize for the confusion caused by a citation error; we will correct the citation from "Liu et al. [4]" to "Liu et al. [3]". In the table below, we compare with more text2shape methods: Text2Shape [5] and CLIP-Forge [38] .
> > | Metrics | CLIP-S | Shapeglot-C | FID |
> > | -------- | -------- | -------- | -------- |
> > | Chen et al.[5]     | 16.29     | 0.14     | 20.21     |
> > | Sanghi et al.[38]     | 26.34     | 0.25     | 21.50     |
> > | Mittal et al.[30]     | 48.92     | 0.46     | 18.45     |
> > | Liu et al.[1]     | 38.88     | **0.50**     | 16.91     |
> > | Ours     | **52.43**     |  **$\geq$ 0.50**    | **16.36**     |
> >
> > We use the best setting for these methods provided by their official code releases. We use shapes without color for [5] and [1]. The table shows that ShapeCrafter outperforms [5], [38], and [30], and is comparable with the concurrent work [1] in text-shape correspondence and shape quality.
> >
> > > **Poor visual quality. Many of the generated shapes (i.e., those in Fig.1, Fig.3, Fig.4) have noticable artifacts, e.g., holes, disconnected or floating parts. Given the visual quality of the demonstrated examples, I would doubt the overall result quality.**
> >
> > A: This is an unfair statement since we do not claim anywhere that our method increases geometry quality. We present a completely new problem setting and choose to use a vector-quantized grid feature decoder that is comparable to other state-of-the-art methods. The visual artifacts are inherited from the implicit shape representation we use. *The table above shows our shape quality is quantitatively comparable with other state-of-the-art approaches and better than previous works in shape quality*. The examples in Figs 1, 3, and 4 in our paper demonstrate our major contribution: generating diverse shapes recursively conditioned on text inputs.
> >
> > > **Missing support for the made claim. Specifically, line 187-188, "As more phrases are added, the probabilities become narrower resulting in more deterministic shapes.". I think this should be a key feature of the method, but somehow it is not validated in the experiments.**
> >
> > A: The table below supports this claim. For a text phrase sequence, we calculate (a) the mean entropy(H) of the per-grid-cell discrete probability distributions $H =-\sum_{i=1}^{g^3}\sum_{k=1}^{K}z_{ik}\log(z_{ik}) /g^3$, and (b) the mean Chamfer Distance(CD) among $8$ randomly sampled shapes. The table below shows how these metrics vary as the number of text phrases increases:
> > | # Phrases | 1 | [2, 4) | [4, 8) | [8, $+\infty$)|
> > | -------- | -------- | -------- | -------- |-------- |
> > | H | 0.423 | 0.341 | 0.294 | 0.267 |
> > | CD | 0.0861 | 0.0698 | 0.0359 | 0.0261 |
> >
> > The table shows that the entropy and the CD decrease with more phrases, i.e. the probability distributions become narrower and the sampled shapes are more deterministic. We thank the reviewer for this suggestion and will add this experiment to the paper.

---

> > > ### Comment · Reviewer_Bwe4 · 2022-08-08
> > > **Reply**
> > >
> > > I appreciate the authors for the clarification and additional experiment results, which definitely help to better understand the method. Though I am still not fully convinced that this task is really that useful in practice (in particularly given the current result quality), I admit that it's an intellectually interesting problem and this work takes a very first step. I feel a bit more positive about this paper after seeing the rebuttal.
> > >
> > > One additional question. Regarding the shape color, I agree that the dataset has poor-quality textures and investigating geometry itself is valuable. But given that previous text2shape method did generate color, I'm just curious how this method would perform if the color is considered? Have the authors tried that? Does it introduce too much complexity that the method fails to handle?

---

> > > > ### Author Response · Authors · 2022-08-08
> > > > **Response to Reviewer Bwe4 - Additional Question**
> > > >
> > > > Thanks to the reviewer for your encouraging response! To answer your question, we didn't try vector-quantized shape representation with appearance, as we think our novel shape representation and architecture for geometry generation/editing is valuable enough. However, the current architecture can be easily extended to colored-shape representation. For example, adding a color decoder along with the occupancy decoder. Our code will be released when the paper is accepted. For future research in text-conditioned shape generation with appearance, our method and code base can be a good baseline to extend from.
> > > >
> > > > Thanks to the reviewer for your suggestions and insightful comments. We revise the paper accordingly as we mentioned. We sincerely hope that the reviewer can re-evaluate the rating in light of the evidence we present to address concerns.

---

> ### Author Response · Authors · 2022-08-08
> **Looking forward to hearing the responses from reviewer Bwe4**
>
> We thank the reviewer for the previous careful reviews and suggestions. We also would like to hear your further comments and suggestions. Please leave a comment and let us know if there are any other questions.

---

### Official Review · Reviewer_e81L · 2022-07-14

**Rating:** 7
**Confidence:** 4
**Soundness:** 3 good
**Presentation:** 3 good
**Contribution:** 3 good

**Summary:**

The paper introduced a novel task, namely recursive text-conditioned shape generation. In this setting, the model needs to generate a 3D shape given the initial text, and subsequently evolves it in multiple steps given more text descriptions.

To achieve this goal, the author built a new dataset (Text2Shape++) which contained many-to-many mappings of phrase sequences and shapes, derived from Text2Shape dataset [5] which only contained one-to-one mappings.

Together with the dataset, the paper also presented a shape generation model that supports generating and refining a shape in multiple iterations. The model is based upon AutoSDF [30]. In the first stage, a VQ-VAE-based shape generation model is trained, and a transformer-based generator is trained on top of the VQ-VAE latent, which can be used to refine a given latent. In the second stage, a recursive feature refinement model based on BERT and 3DCNN is trained directly with text and VQ-VAE latent.


**Questions:**

* The math notations in Section 4.2 is confusing. I wonder what is the $j$ in L178 representing?
* L197 mentioned that the $Z_t$ generated by the recursive language model is send through a transformer before feeding to the 3D decoder. I wonder how important this step is? And how is this implemented? Do you sample from $Z_t$ before feeding the sample into the transformer?

**Limitations:**

Limitations and societal impacts are adequately addressed in the paper.

**Strengths And Weaknesses:**

### Strengths
* The proposed recursive generation task is both novel and useful, which can inspire future research.
* The curated dataset is a valuable and unique addition to the current text-to-shape datasets.
* The new model achieved better generation quality on the proposed dataset compared to previous work, justifying the need of a model tailored for the new task.

### Weaknesses
* The proposed model is trained with strong supervision enabled by the new dataset, as the exact matching between arbitrary phrase sequences and shape sets is available. Building such a dataset is not always feasible for real-world applications with more shape classes and more complex shapes.

---

> ### Author Response · Authors · 2022-07-30
> **Response to Reviewer e81L**
>
> We thank the reviewer for the careful reviews and constructive suggestions. To recap, recursive text-conditioned shape generation is a novel task, which has not been investigated by the community before. We are glad to see all reviewers appreciating our work in the following aspects: (1) the task is novel and useful (e81L, BoQp); (2) the dataset is constructed in an automatic way (Bwe4), which is useful and unique (e81L, Bwe4); (3) the shape set representation as a probability distribution is interesting (Bwe4); (4) the method that supports probabilistic generation is new (Bwe4, BoQp); (5) the method is effective and generates better quality shapes than previous works (e81L); (6) apply the method to a real-world demo (Bwe4); (7) the paper is well-written (BoQp).
>
> In this response, we focus on addressing questions, providing evidence, and addressing factual errors.
>
> ### Answers to the questions
>
> > **The proposed model is trained with strong supervision enabled by the new dataset, as the exact matching between arbitrary phrase sequences and shape sets is available. Building such a dataset is not always feasible for real-world applications with more shape classes and more complex shapes.**
>
> A: In the paper, we propose a *fully automatic way* to construct the dataset, which is a part of our contribution. The method can augment any dataset that has one-to-one text-to-shape correspondences to a dataset with many-to-many correspondences using NLP tools (as reviewer Bwe4 mentioned). This method can also be generalized to build datasets for more shape classes and more complex shapes as well. Given the novelty of our recursive text-conditioned shape generation task, we take the first step of solving it in a supervised manner. We fully agree with the reviewer that future research should investigate how to solve the problem in an unsupervised or self-supervised manner, and we hope that our dataset enables it.
>
>
> > **L197 mentioned that the $Z_t$ generated by the recursive language model is send through a transformer before feeding to the 3D decoder. I wonder how important this step is? And how is this implemented? Do you sample from $Z_t$ before feeding the sample into the transformer?**
>
> A: The table below shows an ablation study of with and without feeding the shape feature to the transformer at the inference step.
>
> | Metric   | CLIP-Similarity | FID |
> | -------- | -------- | -------- |
> | w/o transformer     | 49.29     | 19.83     |
> | Ours     | 52.43     |  16.36     |
>
> The table shows that feeding the shape feature to the transformer leads to higher shape quality. The feature grid is more spatially correlated with the transformer. The higher shape quality also helps the improvement of CLIP-Similarity.
>
> At inference time, we sample from the $Z_t$ before feeding the sample into the transformer. In the next step, we feed the output probability distribution of the transformer to the recursive language model. We will open-source the implementation code when the paper is accepted.
>
> > **The math notations in Section 4.2 is confusing. I wonder what is the $j$ in L178 representing?**
>
> A: Thanks for pointing it out. There are typos in L177-180:
> - L177&L179: $X_j$ --> $X^j$.
> - L177& L180: $Q_j$ --> $Q^j$.
> - L178: $Z_{ik} = \mathbb{I} \[ j = Q_{i} \] $ -->$Z_{ik} =  \mathbb{I} \[ k = Q_{i} \]$. This is a one-hot representation of the latent feature index $Q_i$, which means only the $k=Q_{i}$-th dimension of grid $i$ is $1$.
>
> Thank you for the other suggestions, we will fix typos in the camera-ready version.

---

> > ### Comment · Reviewer_e81L · 2022-08-08
> > **Thanks for the response!**
> >
> > I would like to thank the authors for the clarifications. They have answered all of my questions and concerns. I would like to maintain my rating of accept. My main consideration is that the proposed task is novel and have practical applications.
> >
> > Best,
> > Reviewer e81L

---

### Meta-Review · Area_Chair_LPsB · 2022-08-25

**Recommendation:** Accept
**Confidence:** Less certain

**Metareview:**

This paper presents a recursive text to shape generation system, introduce a new dataset for text-to-shape, and presents good performance of the proposed method. This paper has the potential of inspiring future work.
I encourage the authors to add a discussion (e.g., at limitations or future work) of the necessity of proper metric for evaluation of shape generation. Consolidating this problem can propel the field forward.


**Award:**

No

---

### Decision · Program_Chairs · 2022-09-14

Accept